# Project Proposal - Fine Tuning Methods for Low-resource Languages

**Anton Johansson**
Department of Computer Science,
Tsinghua University
wh24@mails.tsinghua.edu.cn

**Tim Bakkenes**
Department of Computer Science,
Tsinghua University
dim24@mails.tsinghua.edu.cn

**Daniel Wang**
Department of Electronic Engineering,
Tsinghua University
wyl24@mails.tsinghua.edu.cn

## Abstract

The rise of Large Language Models has not been inclusive of all cultures. The models are mostly trained on English texts and culture which makes them underperform in other languages and cultural contexts. By developing a generalizable method for preparing culturally relevant datasets and post-training the Gemma 2 model, this project aims to increase the performance of Gemma 2 for a underrepresented language and showcase how others can do the same to unlock the power of Generative AI in their country and preserve their cultural heritage.

## 1 Introduction

We are participating in the Kaggle competition *Google - Unlock Global Communication with Gemma*, which aims to improve the performance of the Gemma language model for smaller languages as an innovative solution for global communication [7].

### 1.1 Background

The rapid development of Artificial Intelligence (AI) means that an increasing number of people use Large Language Models (LLMs). Since most LLMs are trained primarily on English data, other languages and cultures risk fading away, a concern highlighted by the World Economic Forum[11]. As these biased models become increasingly integrated into society, their potential to amplify existing biases and generate harmful content is intensified, as noted by Torrielli [10]. This emphasizes the importance of developing LLMs that support more than just English language and culture.

Despite some multilingual capabilities, models like Gemma often underperform in non-English languages, showing reduced accuracy and consistency [11].

### 1.2 Importance and Impact

It has been established that Google's Gemma 2 model was primarily trained on English data, leading to reduced performance in other languages and cultures. According to Yogarajan et al. [12], the ideal scenario would be to design LLMs with underrepresented groups in mind from the start. However, even though this might be possible in the future, the bias problem needs to be addressed immediately.

By showcasing methods and technologies whilst developing a fine-tuned version of Gemma 2, other developers will have a roadmap to adapt Gemma 2 for their purposes. Tailoring LLMs to underrepresented languages will empower people from all around the globe to utilize LLMs to solve diverse real-world problems. At the same time, it can be used to preserve the linguistic and cultural identities of local communities and enhance global communication.

However, the applications of tailoring LMMs is not limited to language and culture. The methods proposed could also be used to tailor models for specific business related tasks [6]. Furthermore, the model developed during this project could enable non-English speakers to utilize a LLM.

## 2  Formal Problem Definition

The goal is to fine tune googles' pre-trained Gemma 2 model for other languages than English and showcase and document the methods used to achieve this. In order to do this, a processed dataset used for post-training to fine tune the model, $D'$, has to be crafted from various sources. The Gemma 2 model parameters, $\theta$, and the context supplied via features like Retrieval-Augmented-Generation, $C_{\text{LLM}}$, should be optimized to minimize the loss function that can be expressed with the following equation where $y$ is the expected result and $M_{\text{Gemma2}}(x, C_{\text{LLM}}; \theta)$ is the predicted result of the model given the prompt $x$, the context $C_{\text{LLM}}$ and the model parameters $\theta$.

$$\sum_{(x,y) \in D'} L(y, M_{\text{Gemma2}}(x, C_{\text{LLM}}; \theta))$$

In other words, the goal is to find the optimal model parameters, $\theta'$, and the best context generation to provide the optimal context, $C'_{\text{LLM}}$, that minimizes the loss function. The problem is expressed in the equation below.

$$(\theta', C'_{\text{LLM}}) = \arg \min_{\theta, C_{\text{LLM}}} \sum_{(x,y) \in D'} L(y, M_{\text{Gemma2}}(x, C_{\text{LLM}}; \theta))$$

## 3  Related Work

Recently Google [5] has fine-tuned Gemma 2 for Japan. They used reinforcement learning from human feedback and specialized post-training techniques to enhance performance in Japanese, without impacting the English performance. Furthermore, AI Sweden [1] has developed a Swedish large-scale generative language model to help Swedish organizations build language applications that previously were not possible. To do this, AI Sweden had to source Swedish data to train the model and documented it well which makes their work relevant to this project.

## 4  Proposed Method and Baseline Comparison

In this project, the idea is to craft a sampled dataset comprised of books, articles, forums, and other web sources, similarly to how AI-Sweden did. By using diverse sources and curating the dataset to maintain cultural sensitivity and language-specific nuances, the model can be better tuned. Some examples of data sources include Litteraturbanken [8] that contains books, ArXiv [2] that contains articles and Flashback [4] which is a forum.

Crafting this dataset would enable post-training techniques such as Reinforcement Learning with Human Feedback, similarly to what google did for Gemma 2 for Japan, that can increase the model's performance and understanding of culture. Techniques such as few-shot prompting and retrieval-augmented generation can also be used to improve the performance of Gemma 2. Google AI [6] states that just using just 20 examples of prompts and responses in the target language can help get Gemma to better solve problems in other languages.

Another way to improve performance would be to create a vector database containing embeddings. It could be used for Retrieval-Augmented-Generation to enable the model to generate context-aware responsesAttard [3]. Utilizing vector databases could enable efficient storage and retrieval of context, minimizing token-based API costs [3]. Other methods to increase performance such as Low-Rank Adaptation and different methods to generate the embeddings based on data availability will also be explored.

The model fined tuned during the course of this project will be compared against models like Gemma 2 with no task-specific fine-tuning, as well as GPT-SW3 [9]. The comparison will focus on translation accuracy, cultural relevance and knowledge, and bias minimization.

# 5   References

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
