# OpenReview forum: "Fine Tuning Methods for Low-resource Language"
_tsinghua.edu.cn/THU/2024/Fall/AML — THU 2024 Fall AML Submission_

### Official Review · ~Ziyi_Liu9 · 2024-11-07
**Good**

**Rating:** 9
**Confidence:** 4

**Review:**

The problem formulation and background are clearly articulated. However, a more detailed summary of related work would enhance the understanding of the current landscape and the unique contributions of this project.

---

### Official Review · ~Jia-Nuo_Liew1 · 2024-11-08
**Clear Proposal**

**Rating:** 9
**Confidence:** 5

**Review:**

The proposal provides a solid foundation for the project with clear motivation, a well-defined problem, relevant related work and structured methodology.

Background: Thourough, explaning the risk of language loss due to English-dominated LLMs and the need to counteract cultural bias.It effectively highlights the larger impact of the project.
Definition: The problem is defined mathematically with clear variables and symbols. The formalization is clear.
Related Works: Offers relevant examples and clear context for research. Though more critical analysis could be done.
Proposed Method: Outlines a thoughtful approach to data collection. Provides a clear plan to incorporate diverse data sources.

---

### Official Review · ~Yunghwei_Lai1 · 2024-11-08

**Rating:** 7
**Confidence:** 4

**Review:**

While the approach is well thought out, there are few specifics on how each step will be implemented. For instance, it would be helpful to know more about the quantity and diversity of data to be sampled from each source and the criteria for evaluating cultural sensitivity in data curation.
The project methodology outlines a solid approach and mentions several advanced techniques (such as RLHF, few-shot prompting, and retrieval-augmented generation), but it lacks specific details on how these methods will be implemented in your project.

Insufficient related background survey in the related work part.

---

### Official Review · ~Zijun_Liu2 · 2024-11-08
**Review and Feedback**

**Rating:** 8
**Confidence:** 4

**Review:**

## Overview
This project aims to address cultural and linguistic biases in Large Language Models (LLMs) by developing methodologies to fine-tune Gemma 2 for underrepresented languages, fostering global inclusivity in AI usage. The authors outline steps for creating culturally sensitive datasets, and employing techniques like Reinforcement Learning with Human Feedback (RLHF) and Retrieval-Augmented Generation.

## Suggestions and Comments
The focus on culturally sensitive data highlights a rigorous plan and could be quite contributory. However, there is a need to compare to other methods as baselines, except untuned Gemma 2, to show significance of collecting such datasets. Besides GPT-SW3, another data collection method, I would recommend the authors to look into the work of \[1\], which does not collect extra data but goes for distillation. Comparison between these methods would show the necessity of collecting extra data.

\[1\] Enhancing Multilingual Capabilities of Large Language Models through Self-Distillation from Resource-Rich Languages. ACL 2024.

---

### Official Review · ~Jinsong_Xiao1 · 2024-11-09
**review for peoposal 35**

**Rating:** 8
**Confidence:** 3

**Review:**

This proposal aims to improve the performance of Google’s Gemma 2 language model for low-resource languages by developing methods for dataset preparation and model fine-tuning. The whole work has good meaning for mankind.

Strengths:

- The project objectives are well-defined, with a focus on cultural sensitivity and inclusivity.

- A successful outcome could serve as a valuable blueprint for adapting LLMs to a range of low-resource languages, preserving linguistic diversity.

Suggestions: More detailed description of related work and evaluation scheme

---

### Official Review · ~Zou_Dongchen1 · 2024-11-10
**Review of this paper**

**Rating:** 7
**Confidence:** 4

**Review:**

This proposal focuses on the social impact of LLM.LLM is mostly trained based on English, making it underperform on low-resource language tasks. English-based trained models may also amplify biases in other cultural environments. Therefore, I think the authors' idea of enhancing LLM's ability on low-resource languages to help preserve the cultural heritage of each country is very meaningful.
The downside of this proposal is that the relevant details are not complete enough. For example, in the problem definition section, the author only gives a very broad optimization problem. This problem basically holds true for any LLM task and does not reflect the uniqueness of low-resource languages.

---

### Official Review · ~Renrui_Tian1 · 2024-11-11
**Promising Topic yet Challenges to Improve**

**Rating:** 8
**Confidence:** 3

**Review:**

**Strengths**:
* **Relevant Problem**: The project identifies a crucial area in LLM development—reducing bias and improving support for non-English languages, which has both academic and social relevance.
* **Methodological Innovation**: Leveraging techniques such as RLHF and vector databases for optimized context generation indicates an innovative approach.

**Weaknesses**:
* Dataset Challenges: The proposal could benefit from elaboration on potential limitations regarding data availability and quality for specific languages, as sourcing culturally rich and representative datasets may prove challenging.

*Overall*: This project has the potential to meaningfully advance the inclusivity and adaptability of LLMs in diverse linguistic and cultural contexts. Further details on data challenges would enhance the proposal’s feasibility.

---

### Official Review · ~Ziyad_Fawzy1 · 2024-11-11
**Fine Tuning Methods for Low-resource Language**

**Rating:** 8
**Confidence:** 5

**Review:**

The authors highlight two separate issues: LLMs' underperformance in low resource languages, and cultural biases in LLM responses. The authors suggest that training or fine-tuning an LLM on a corpus of underrepresented languages could both improve its linguistic abilities for low-resource languages and reduce the cultural/social bias in its responses.

The issues highlighted by the authors are significant and meaningful. The approach and the focus of the paper could benefit from narrowing the scope of the issues the authors plan to address. While a corpus of low-resource languages might contain relevant cultural context that is underrepresented the English training data, it is not clear that this cultural or social context will generalize in the LLM's responses in English or other languages (the authors should consider this limitation as it relates to their ability to evaluate their model). Additionally, using a general dataset to target a specific problem such as social bias might not be the most effective approach. Any general dataset used will also come with its own set of limitations and social biases.

The authors also highlighted the issue of LLM underperformance in generating low-resource languages. The authors should consider whether the languages they are targeting have existing datasets that can be used for fine-tuning. Compiling a new dataset is also possible, but the amount of data needed to meaningfully target the language generation task and the time needed to collect such data should be considered.

---

### Official Review · ~Cheng_Gao2 · 2024-11-11
**Review for Fine Tuning Methods for Low-resource Language**

**Rating:** 8
**Confidence:** 4

**Review:**

Strengths:

- The issues highlighted in the proposal is closely related to society, holding significant practical value.
- The author has introduced methods such as RAG, LoRA, and Vector base to enhance model performance.

Weaknesses:

- The author’s proposed approach is “crafting a sampled dataset comprised of books, articles, forums, and other web sources”. I am concerned that it may not be fully applicable to “Low-resource Languages.” I thought “Low-resource Languages” means they may not have sufficient data available from these common sources for model trainning. In other words, the proposed method may not be transferable to languages with very limited online resources.
- Although RAG, LoRA, and Vector base are indeed common methods for improving model performance or reducing training complexity, they may lack innovation when it comes to training models for Low-resource Languages. I would recommend exploring alternative approaches, such as generating synthetic data or utilizing small translation datasets to attempt transferring the model’s capabilities from English to Low-resource Languages.

---

### Official Review · ~Chenxi_Hu4 · 2024-11-11
**A Promising Approach for Underrepresented Languages**

**Rating:** 8
**Confidence:** 3

**Review:**

This proposal presents a well-structured approach to enhancing the multilingual capacity of the Gemma 2 language model, focusing on underrepresented languages. The proposed method employs few-shot prompting and retrieval-augmented generation, which are promising strategies for achieving better contextual understanding. However, details on specific metrics to assess "cultural relevance and knowledge" could strengthen the evaluation framework. Overall, this project has the potential to make a valuable contribution to the inclusivity and cultural diversity of LLMs.

---

### Official Review · ~Grace_Xin-Yue_Yi1 · 2024-11-11

**Rating:** 8
**Confidence:** 3

**Review:**

The proposal clearly explains the problem and its relevance, making a strong case for the importance and impact of the research. It also includes a clear formalization of the problem. The methodology is well-structured, describing a plan to source culturally relevant data and implement post-training techniques. However, the proposal could benefit from including evaluation metrics and a more thorough analysis of related works.

---

### Official Review · ~Han-Xi_Zhu1 · 2024-11-12
**Review for Fine Tuning Methods for Low-resource Language**

**Rating:** 9
**Confidence:** 4

**Review:**

The project is commendable for its focus on inclusivity and its potential to enhance LLM performance for underrepresented languages.\
## Strength
1. The project propose a significant task in AI research by focusing on underrepresented languages and cultures, which is crucial for global inclusivity and cultural preservation.
2. The project has practical applications in improving global communication and could serve as a model for other languages, potentially influencing the development of LLMs.
3. The project is well writen.


## Weakness
1. The success of the project heavily depends on the quality and representation of the dataset. Especially the authors try to deal with the Low-resource Language. Where can we get enough training data?